# A Potential Adhesin/Invasin STM0306 Participates in Host Cell Inflammation Induced by *Salmonella enterica* Serovar Typhimurium

**DOI:** 10.3390/ijms24098170

**Published:** 2023-05-03

**Authors:** Chong Ling, Shujie Liang, Yan Li, Qingyun Cao, Hui Ye, Changming Zhang, Zemin Dong, Dingyuan Feng, Weiwei Wang, Jianjun Zuo

**Affiliations:** Guangdong Provincial Key Laboratory of Animal Nutrition Control, College of Animal Science, South China Agricultural University, Guangzhou 510642, China

**Keywords:** *S.* Typhimurium, STM0306, adhesin, cellular inflammation, infection

## Abstract

*Salmonella enterica* serovar typhimurium (*S.* Typhimurium) is a common Gram-negative foodborne pathogenic bacterium that causes gastrointestinal disease in humans and animals. It is well known that adhesins and invasins play crucial roles in the infection mechanism of *S.* Typhimurium. *S.* Typhimurium STM0306 has been denoted as a putative protein and its functions have rarely been reported. In this study, we constructed the *STM0306* gene mutant strain of *S.* Typhimurium and purified the recombinant STM0306 from *Escherichia coli*. Deletion of the *STM0306* gene resulted in reduced adhesion and invasion of *S.* Typhimurium to IPEC-J2, Caco-2, and RAW264.7 cells. In addition, STM0306 could bind to intestinal epithelial cells and induced F-actin modulation in IPEC-J2 cells. Furthermore, we found that STM0306 activated the nuclear factor kappa B (NF-κB) signaling pathway and increased the mRNA expression of pro-inflammatory cytokines such as IL-1β, TNF-α, as well as chemokine CXCL2, thus resulting in cellular inflammation in host cells. In vivo, the deletion of the *STM0306* gene led to reduced pathogenicity of *S.* Typhimurium, as evidenced by lower fecal bacterial counts and reduced body weight loss in *S.* Typhimurium infected mice. In conclusion, the STM0306 of *S.* Typhimurium is an important adhesin/invasin involved in the pathogenic process and cellular inflammation of the host.

## 1. Introduction

*Salmonella* is a very large family, containing more than 2500 serotypes, which are distinguished by their different antigens. According to the specific antigenic structures, *Salmonella* can be divided into two categories: *Salmonella bongori* and *Salmonella enterica* [1], the latter containing at least 90% of *Salmonella* serotypes.

Worldwide, more than 153 million people are infected with *Salmonella* every year. At least 50,000 people die from *Salmonella* infections, of which about 52% are foodborne [2]. Foodborne diseases caused by *Salmonella* are a serious public health problem, and animal food is the main source of human salmonellosis [3]. In humans, approximately 50% of salmonellosis is caused by *S.* Typhimurium [4]. *S.* Typhimurium is one of the most isolated *Salmonella* worldwide [5], and as a foodborne and zoonotic pathogen that can infect mice, poultry, pigs and humans, *S.* Typhimurium has a significant impact on public health security.

After reaching the small intestine, *Salmonella* contacts with intestinal epithelial cells and adheres to the cell surface through various fimbriae and outer membrane proteins distributed on the thallus surface [6], moreover, the flagellum is also involved in *Salmonella* adhesion [7]. This is followed by internalization into host cells under the action of effectors encoded by *Salmonella* pathogenicity island-1 (SPI-1) and invasins such as Rck and PagN [8]. SPI-1 encodes the type III secretion system-1 (T3SS-1), which injects effectors into the host cells, leading to bacterial internalization (the “trigger” mechanism) [9]. *Salmonella* can also invade cells through a “zipper” mechanism mediated by invasins. The “zipper” mechanism is a cellular signaling and bacterial internalization mechanism that results from the interaction of invasins with host cell receptors [10].

Adhesion to host cells is known to be a crucial step in the pathogenesis of *Salmonella*. The process of adhesion is realized by the interaction between the receptor on host cells and the adhesion structures present on the bacterial surface, such as FimH (type I fimbriae), SafD (Saf fimbriae), OmpV, and SiiE [11,12,13,14,15]. Notably, in addition to their role in mediating bacterial adhesion, these adhesion structures also elicit host cell inflammatory and immune responses [16,17]. Thus, an adhesin is an important bacteria virulence factor that can enhance the bacteria pathogenicity by promoting adhesion to the host mucosa surface and triggering the body’s immune response.

STM0306 has been defined as a hypothetical adhesin/invasin by GenBank (National Center for Biotechnology Information). A previous study suggested that *S.* Typhimurium STM0306 is a homolog of T2544, which facilitates *S*. Typhi adhesion to host cells and plays an adhesive role by binding to laminin on the host cell membrane, however, STM0306 was not found to be associated with the adhesion function of *S.* Typhimurium [18]. In this paper, we report that STM0306 is highly homologous to PagN, which is an adhesin and an invasin regulated by the PhoP-PhoQ two-component regulatory system [19]. To further explore the relationship between STM0306 and bacterial adhesion, we conducted a comprehensive study to explore the possible role of STM0306 in bacterial adhesion and its further functions in immune responses against host cells. The results of our research may provide more information on the role of STM0306 as an adhesin/invasin in the adhesion and pathogenesis of *S.* Typhimurium.

## 2. Results

### 2.1. STM0306 Is a Homolog of PagN

Sequence alignment using CLUSTALW (https://www.genome.jp/tools-bin/clustalw, accessed on 12 March 2022) showed that the similarities between T2544 (*S*. Typhi, NCBI Protein ID: AAO70128.1), STM0306 (*S.* Typhimurium, NCBI Protein ID: NP_459304.1), and PagN (*S. bongori*, NCBI Protein ID: WP_024143149.1) are high (Figure 1). The similarity between STM0306 and T2544 reached more than 98% and the similarity between STM0306 and PagN was 83%. The identity between them reached 82%. 

To get a clear idea of the function of STM0306 in the *S.* Typhimurium, we constructed the *STM0306* mutant strain (Δ*0306*) using the λ-red recombinase system. PCR amplification, using the specific primer of *STM0306,* indicated that the *STM0306* gene in the mutant strain was deleted (Figure 2A). We also constructed a complemented *STM0306* strain (CΔ*0306*) based on the mutant strain, using the high copy cloning plasmid pBR322 and restriction endonuclease (Himd III and EcoR V). The expression of the *STM0306* gene in each strain was verified by qRT-PCR (Figure 2B). The results showed that the *STM0306* gene was overexpressed in the CΔ*0306* strain. Furthermore, we expressed the *STM0306* gene in *Escherichia coli* using expression plasmid pET-32a and restriction endonuclease (XhoI and BamH I) and purified recombinant STM0306 protein (r0306) from the supernatant of the lysed bacteria. SDS-PAGE showed that the heteroproteins in the lysate had been removed after purification (Figure 2C).

### 2.2. STM0306 Is Involved in Adhesion and Invasion of S. Typhimurium

The amino acid sequences of T2544, STM0306, and PagN are similar, and both T2544 and PagN are related to bacterial adhesion and invasion. To explore the effect of *STM0306* gene deletion on the adhesion and invasion ability of *S.* Typhimurium, we performed a standard adhesion and invasion assay by using the IPEC-J2, Caco-2, and RAW264.7 cell lines. The IPEC-J2 cell line mimics humans more closely than other cell lines of non-human origin, which is important in studies of zoonotic infections [20] and is therefore an ideal model for studying bacterial interactions with intestinal cells [21]. The Caco-2 cell line is morphologically and functionally similar to small intestinal epithelial cells [22] and is widely used to study the adhesion of pathogenic bacteria. The RAW264.7 cell line is a murine monocyte-macrophage leukemia cell line [23] and is commonly used to study *Salmonella* survival and growth in macrophages [24]. 

The numbers of adhering and invading bacteria of the Δ*0306* strain were reduced (*p* < 0.01) compared with the WT and CΔ*0306* strains, however, the ratio of invaded bacteria to adhered bacteria showed that bacterial invasiveness was not impaired (*p* > 0.05) (Figure 3A), indicating that the deficiency of the *STM0306* gene attenuates the adhesion ability of *S.* Typhimurium but not invasion to IPEC-J2 cells. In the mechanism of *Salmonella* infection, invasion is the next step after adhesion, and the standard invasion assay cannot reduce the disparities in bacterial adhesion. Therefore, the reduction in the number of invading bacteria seemed to be attributed to the decrease in adhered bacteria. Analogous adhesion results were obtained using Caco-2 and RAW264.7 cells as the object, however, we observed a decrease (*p* < 0.05) in invasiveness of the mutant strain to Caco-2 and RAW264.7 cells (Figure 3B,C).

*Salmonella* is able to survive in macrophages [25]. Once it enters macrophages, *Salmonella* can form *Salmonella*-containing vacuoles (SCVs), which provide a site for its proliferation and expression of a variety of virulence genes, including *pagN* [26]. Therefore, we investigated whether there was any change in the proliferation of *S.* Typhimurium in macrophages after deletion of the *STM0306* gene. The results showed a reduction in the number of bacteria (*p* < 0.05) of the Δ*0306* strain in RAW264.7 cells compared with the WT strain (Figure 3D). However, relative to the number of invading bacteria, there was no significant change (*p* > 0.05) in the intracellular proliferation of the mutant strain.

The invasiveness of *S.* Typhimurium was restored and enhanced by *STM0306* complement in RAW264.7 cells, which could be due to the overexpression of the *STM0306* gene in the CΔ*0306* strain, and might also be responsible for changes in the expression of other virulence genes. To confirm this speculation, we then performed qRT-PCR. As shown in Figure 3E, the expression of multiple virulence genes such as *csgA*, *hilA*, *hilD*, *sopB*, *sopD*, *sopE*, and *mipA* in the CΔ*0306* strain were higher (*p* < 0.05) than in the WT bacteria, implying that the high expression of *STM0306* up-regulated the expression of other invasion and virulence genes. In addition, we found an increase (*p* < 0.05) in the expression of *sopD* in the Δ*0306* strain compared with the WT strain.

### 2.3. Recombinant STM0306 Reduces the Adhesion of S. Typhimurium through Binding to Host Cells

To further investigate the functions of the *STM0306* gene, we expressed *STM0306* (induced by IPTG) using the expression vector pET-32a and purified recombinant STM0306 (r0306) by Ni-NTA affinity chromatography, as mentioned above. An ELISA-based method was used to probe whether STM0306 has the ability to bind to host intestinal epithelial cells. As shown in Figure 4A, the optical density value of the r0306 treatment group was dose-dependently higher (*p* < 0.01) than that of the control group. This suggests that r0306 has the function of adhering to IPEC-J2 cells.

Our results above strongly indicate that STM0306 is a potential adhesin of *S.* Typhimurium. Adhesins can bind to corresponding receptors on the cell membrane, which are also required for bacterial adhesion. Therefore, it is possible to inhibit bacterial adhesion by neutralizing the extracellular receptors using recombinant adhesins. We hypothesized that r0306 pretreatment could affect the adhesion of *S.* Typhimurium to intestinal epithelial cells by adhering to the specific receptors on the cell surface. Thus, IPEC-J2 cells were pretreated with r0306 and then infected with *S.* Typhimurium followed by a standard adhesion assay. As expected, there was a dose-dependent decrease in adherent bacteria in the r0306 preincubation group compared with the control group (Figure 4B).

To explore whether the reduction in bacterial adhesion caused by r0306 preincubation was due to the reduction in cell viability, CCK-8 reagent was used to detect the cell viability after r0306 treatment. As shown in Figure 4C, there was no distinction (*p* > 0.05) in the cell viability between the r0306 pretreatment group and the control group at all doses. Furthermore, the adhesion of *S.* Typhimurium was observed by Wright–Giemsa staining under the same treatment conditions. As shown in Figure 4D, the amount of bacterial adhesion in the r0306 pretreatment group was less than that in the control group. In order to examine this change more intuitively, we made an *S.* Typhimurium strain labeled with spontaneous green fluorescence, using plasmid pUC19-EGFP to observe the fluorescence distribution under a fluorescence microscope. The result showed that the intensity of green fluorescence in the r0306 pretreatment group was less (*p* < 0.01) than that in the control group (Figure 4E).

The above results suggest that STM0306 could serve as an adhesin/invasin that inhibits the adhesion of *S.* Typhimurium by binding to IPEC-J2 cells in a manner independent of cell viability.

*Salmonella* can invade cells through a “zipper” mechanism that is mediated by invasins such as Rck, HlyE, and PagN [27,28,29], which involve the modulation of the actin filaments [30]. Thus, we investigated whether STM0306 has the ability to induce actin regulation. As shown in Figure 5, the cytoskeleton of IPEC-J2 cells was deformed and rearranged after r0306 treatment, and the tight junction between cells was looser than that in the solvent control group, indicating that STM0306 could lead to the actin modulation of IPEC-J2 cells.

### 2.4. STM0306 Participates in Inducing Host Cell Inflammation

Generally, bacterial adhesins are associated with cellular inflammation and immunity. To examine the effect of STM0306 adhesion on the expression of inflammatory factors in host cells, IPEC-J2 cells were treated with r0306 to determine the change in the relative mRNA expression of inflammatory factors. As shown in Figure 6A, the expression of *TNF-α* and *IL-1β* was up-regulated in the r0306 treatment group (*p* < 0.01) compared with the control group. In terms of inflammatory chemokines, r0306 treatment could lead to a dose-dependent increase (*p* < 0.01) in the expression of *CXCL2*, an important target gene of the nuclear factor kappa B (NF-κB) signaling pathway [31], indicating that r0306 may mediate the expression of inflammatory factors by activating the NF-κB signaling pathway. In addition, a dose-dependent decrease in the mRNA level of *occludin* was observed in the r0306 treatment group.

To investigate whether the r0306-mediated expression of inflammatory factors was caused by lipopolysaccharide (LPS) in the purified protein, we determined the LPS contamination in the protein solution. The results showed that the LPS level in the purified protein was 1.73 pg/μg, which did not induce inflammatory responses in IPEC-J2 cells.

Western blotting verified a dose-dependent increase in the expression of phosphorylation of p65 (p-p65) (Figure 6B) in IPEC-J2 cells treated with r0306.

The above findings indicated that r0306 can induce inflammatory responses of IPEC-J2 cells via activation of the NF-κB signaling pathway.

### 2.5. STM0306 Deletion Reduces the Pathogenicity of S. Typhimurium in Mice

*S.* Typhimurium infection can cause symptoms of typhoid fever in mice, we thus investigated the effect of *STM0306* gene deletion on pathogenicity of *S.* Typhimurium in vivo. The results revealed a decrease (*p* < 0.01) in the body weight of the WT strain group compared with the control group after oral administration of *S.* Typhimurium (Figure 7A). In addition, the weight loss in mice infected with the Δ*0306* strain was less (*p* < 0.05) than that in the WT strain-infected mice. We also found that the number of bacteria in the feces of Δ*0306*-infected mice was less (*p* < 0.05) than that for WT-infected mice one day after infection (Figure 7B). Furthermore, the mortality rate in mice infected with Δ*0306* was lower than that in mice infected with either the WT or CΔ*0306* strain (Figure 7C).

Overall, deletion of the *STM0306* gene reduces the pathogenicity of *S.* Typhimurium to mice.

## 3. Discussion

*Salmonella* initiates the infection by attaching to the intestinal mucosa and then invading epithelial cells. This process depends on the action of flagellum, fimbriae, adhesins, and invasins. The flagellum is the motility system of *Salmonella*, however, recent studies have shown that flagella also play an essential role in infection processes, such as adhesion [32]. Fimbriae are filamentous protein structures existing on the surface of bacterial cells, which can mediate bacterial adhesion to host cells. In addition, the cell surface of *Salmonella* also contains a variety of outer membrane proteins with adhesive properties, namely adhesins, which have functions similar to fimbriae. Invasins located on the bacterial cell surface also contribute to mediating the pathogenicity of *Salmonella*. In general, *Salmonella* contain three known invasins of *Salmonella*: Rck, PagN, and Hyle. Among them, Rck and PagN not only have the ability to invade host cells, but also mediate bacterial adhesion to host cells.

During the past two decades, extensive research has been conducted to delete and purify certain typical adhesins and invasins of *Salmonella* to investigate the functions of these structures in vitro. However, it is reasonable to speculate that additional adhesins governing *Salmonella* adhesion remain to be discovered. In this study, we found that STM0306 of *S.* Typhimurium and PagN of *S*. *bangori* are highly homologous. The amino acid sequences of PagN are highly similar among many serotypes of *Salmonella* [19]. To identify the functions of STM0306 in *S.* Typhimurium, we conducted the *STM0306* gene mutant and complemented strains, the qRT-PCR verification showed that the *STM0306* gene was barely expressed in the mutant strain and overexpressed in the complemented strain. However, a clean complementation strain was lacking in this study, and the construction of a complementation strain with the same expression as the wild-type strain might better explain the functions of the *STM0306* gene. Nevertheless, in this article, the overexpressed strain restored the adhesion and invasion phenotypes of the *STM0306* mutant strain, and even slightly enhanced them in some aspects, which could confirm the functional specificity of the *STM0306* gene to some extent.

The adhesion and invasion results showed that the deletion of *STM0306* gene caused a reduction in the adhesion and invasion (except IPEC-J2 cells) abilities of *S.* Typhimurium, which were restored after a complement of the *STM0306* gene, confirming a role of STM0306 in contributing to mediating the adhesion and invasion of *S.* Typhimurium. This was not consistent with the study of Ghosh et al. [18], who found that mutant *S*. Typhi lacking the STM0306 homolog T2544 showed completely blocked adhesion to HT-29 cells, but an STM0306 mutant did not. The discrepancy might be related to different cell types, as with the differences in mutant bacterial invasion on IPEC-J2 cells and other cells in our results. Notably, this study showed that the deletion of the *STM0306* gene did not completely block the adhesion and invasion of *S.* Typhimurium to host cells. However, it has been reported that deletion of the *pagN* gene results in an almost complete loss of the capacity of bacteria to adhere to and invade HT-29 cells [19]. Furthermore, Lambert et al. [29] found that the invasiveness of the *pagN* mutant to different cells was distinct, specifically, the invasiveness of the *pagN* mutant strain to CHO-K1 cells was reduced to a similar degree as in the results of our study, although the *pagN* mutation elicited little change in bacterial invasiveness to pgsA745 cells. It was established that PagN can interact with heparan sulfate proteoglycan (HSPG), a crucial component of the mammalian cell membrane [33]. Therefore, the diversity of invasiveness of the *pagN* mutant strain to different cells may be related to the corresponding variation in HSPG content, which might not be the determinant of bacterial invasiveness. In support of this view, Barilleau et al. [34] suggested that cell surface HSPG content is not associated with PagN-mediated invasiveness of bacteria (recombinant *Escherichia coli* HB101), as supported by the fact that HT-29 cells with the lowest HSPG content showed the maximum number of invasive bacteria, while pgsA745 and Caco-2 cells had different numbers of invasive bacteria even though their HSPG contents was similar. Remarkably, PagN-mediated bacterial internalization requires the participation of β1 integrin, in addition, PagN possibly mediate bacterial internalization through the joint action of HSPG and β1 integrin [34]. The integrins are a large family of adhesion receptors consisting of two subunits (α and β subunits). β1 integrin is one of the most common integrin subunits that binds to different α subunits, such as laminin-binding integrins (α1β1, α2β1, α3β1, α6β1, and α7β1) and collagen-binding integrins (α1β1, α2β1, α3β1, α10β1, and α11β1) [35]. However, the another α subunit of integrin that binds to the STM0306 or PagN is still unknown. Indeed, the study by Wu et al. [36] revealed that different amino acid residues at 49 and 109 of *pagN* alleles could lead to different pathogenic properties, such as adhesion and invasion, which is consistent with our results that the amino acid residues at 49 and 109 among T2544, STM0306, and PagN are not identical. However, the differences in treatment conditions may also be responsible for different adhesion/invasion results. Using the same methods and conditions as the above researchers as controls would better explain the differences in the results of the mutant strains on different cell types. Accordingly, the degree of influence of the *STM0306* gene deletion on the invasiveness of *S.* Typhimurium to distinct cells might be related to the different contents of receptors (HSPG, β1 integrin and related α integrin) on the cell surface, treatment conditions, and allelic variation in virulence factors.

After entering SCV, the PhoP/Q two-component system of *S.* Typhimurium is rapidly activated by environmental signals, such as low magnesium, cationic peptides, and antimicrobial peptides, to promote the expression of *pagN* (which reaches a maximum in SCV) and a large number of other virulence genes [37,38,39]. Herein, we evidenced that the *STM0306* gene does not participate in the intracellular proliferation of *S.* Typhimurium.

Through the invasion assay, we detected that the *STM0306* complement made up for the deficiency of the invasion ability of *S.* Typhimurium. Interestingly, the *STM0306*-complemented strain induced an increase in the invasiveness to a level greater than that induced by the WT strain. This result could be due to a change in the expression of other virulence or invasion genes in addition to the overexpression of *STM0306* in the complemented strain. Indeed, the qRT-PCR results confirmed that the relative expression of virulence regulatory genes *csgA*, *hilA*, and *hilD* coupled with invasive protein genes *sopB*, *sopD*, *sopE,* and *mipA* were significantly higher in the complemented *STM0306* strain than in the WT strain. CsgA is the major subunit protein of curli pili, which are participating in the adhesion and invasion of bacteria to host cells [40]. HilA and HilD are pivotal transcriptional regulators of *S.* Typhimurium [41]. HilA is located in SPI-1 and serves as a central regulator and transcriptional activator of SPI-1, while HilD is the core part of the HilC-RtsA-HilD feed-forward regulatory loop controlling the regulatory network of HilA [42]. Each activating factor in the HilC-RtsA-HilD loop can combine with the *hilA* promoter to activate the expression of the *hilA* gene, and the HilA protein can also induce its own expression [42,43,44]. SopB, SopD, and SopE belong to the Sops effector protein family, and are involved in different stages of polymorphonuclear leukocyte influx and cytoskeleton rearrangement [45,46,47]. MipA, a homolog of the *Salmonella* Rck invasin, is also one of the most important invasins of *S.* Typhimurium. Therefore, the increased expression of *csgA*, *hilA*, *hilD*, *sopB*, *sopD*, *sopE*, and *mipA* genes of *S.* Typhimurium following *STM0306* complementation, implied a synergistic relationship between *STM0306* and other invasion genes. This could account for the observed enhancements in the invasiveness of the complemented *STM0306* strain to RAW264.7 cells relative to the WT strain. In addition, we did not observe an impaired invasiveness of the *STM0306* mutant to IPEC-J2 cells that could be attributed to the up-regulation of *sopD*.

To further explore the function of the *STM0306* gene and its encoded protein, we successfully expressed and purified recombinant STM0306 (r0306) using the expression vector pET-32a and *Escherichia coli* BL21 (DE3). Invasins mediate bacteria internalization through the “zipper” mechanism. After attachment, the bacteria closely bind to the host cell membrane, and initiate a minor cytoskeletal protein rearrangement through specific contact between the invasin and host cell surface receptors. To investigate whether r0306 induces cytoskeleton rearrangement, IPEC-J2 cells were pretreated with r0306 and then stained with FITC-labeled phalloidin. The cytoskeleton is the protein fiber network structure in eukaryotic cells and composed of microtubules, microfilaments, and intermediate fibers [48]. Phalloidin is a kind of polypeptide substance, which can selectively and tightly bind to F-actin of the cytoskeleton [49]. In this study, microscopic examination showed that r0306 treatment could lead to local deformation and rearrangement of the cytoskeleton of IPEC-J2 cells and loosen the tight connection between cells, suggesting a role of STM0306 in mediating the destructive effect of *S.* Typhimurium on the host cytoskeletal structure.

Fimbriae and adhesins have multiple functions in the initial stage of bacterial infection. In addition to the role in intestinal colonization, adhesins can also induce the tissue tropism for Peyer’s patches [50]. Furthermore, adhesins participate in the induction of host cell inflammatory responses by contacting with host epithelial cells. For example, Uchiya et al. [16] showed that FimH adhesin induced the expression of the inflammatory factors IL-1β, IL-6, and TNF-α by activating the NF-κB signaling pathway and the mitogen-activated protein kinase (MAPK) signaling pathway in macrophages. Similarly, the present study revealed inflammatory responses in IPEC-J2 cells caused by r0306 treatment. However, the inflammatory responses induced by r0306 treatment were not as obvious as those induced by FimH reported by Uchiya et al. [16], probably because STM0306 is not the main adhesin of *S.* Typhimurium.

Apart from inflammatory responses, we also found that r0306 treatment reduced the expression of *occludin* at the mRNA level in host cells, which was deduced to clarify the loosened tight junction within IPEC-J2 cells treated with r0306, as revealed by cytoskeleton staining.

In vitro, STM0306 adhered to host intestinal epithelial cells and then induced host cell inflammation along with disruption of tight junction proteins, which were possibly detrimental to the body weight and survival of host. Indeed, we observed that the *STM0306* deletion reduced the number of intestinal bacteria and alleviated the weight loss and death of the mice.

## 4. Materials and Methods

### 4.1. Cell Lines, Bacterial Strains, and Mice

IPEC-J2, Caco-2, and RAW264.7 cell lines were cultured under standard conditions with Dulbecco’s modified Eagle’s medium (DMEM) (Hyclone, Utah, USA) supplemented with 4 mM L-glutamine, 1 mM sodium pyruvate, and 10% (*v*/*v*) fetal bovine serum (FBS, Gibco, New York, NY, USA), grown at 37 °C in 5% CO_2_. BALB/c mice were obtained from Guangdong experimental animal center (Guangzhou, China). All bacterial strains and plasmids used in this study are listed in Table 1.

### 4.2. Construction of Mutant Strain and Complemented Strain of S. Typhimurium

The *STM0306* gene of the *S.* Typhimurium wild-type (WT) strain was deleted through the λ-red recombination system. Firstly, the target DNA fragment was amplified from the plasmid pKD4 and transformed into *S.* Typhimurium. Afterwards, homologous recombination was performed under the action of the recombinant enzyme expressed by the plasmids pKD46 and pCP20, followed by replacement of the *STM0306* gene with its homologous sequence to obtain the *STM0306* mutant strain (Δ*0306*). Finally, the recombinant plasmid pBR322-*STM0306* (Himd III and EcoR V) was constructed and transformed into the Δ*0306* strain to obtain the *STM0306* complement strain (CΔ*0306*). The primer sequences used for molecular cloning are listed in Appendix A.

### 4.3. Expression and Purification of Recombinant STM0306 Protein (r0306)

To obtain the recombinant protein, recombinant plasmid pET-32a-*STM0306* (XhoI and BamH I) was constructed and expressed in *E. coli* BL21 (DE3) treated with 1 mM isopropyl-beta-D-thiogalactopyranoside (IPTG) (Sangon, Shanghai, China), at an OD_600_ of 0.6, for 5 h. The bacteria were then lysed with 1 mg/mL lysozyme (Sangon, Shanghai, China), and the resulting r0306 was purified using the His-Tag protein purification kit (Beyotime, Shanghai, China).

### 4.4. Adhesion and Invasion Assay

IPEC-J2 and Caco-2 cells were cultured in 12-well culture plates at 37 °C with 5% CO_2_ overnight, followed by infection with the *S.* Typhimurium at a multiplicity of infection (MOI) of 100 bacteria per cell, for 1 h. The cells were washed with PBS to eliminate non-adhered bacteria and then treated with PBS containing 1% Triton X-100 (*v*/*v*) for 30 min. For the invasion assay, cells were treated with gentamicin (100 μg/mL) for 1 h to kill adhered bacteria. Thereafter, cells were washed with PBS and incubated with PBS containing 1% Triton X-100 (*v*/*v*), referring to the adhesion assay, followed by enumeration of *S.* Typhimurium using the spread plate method. Bacterial invasiveness was depicted as a percentage of adhered bacteria. For the r0306 adhesion assay, IPEC-J2 cells were pretreated with r0306 or eluent solvent for 1 h before bacterial infection.

### 4.5. Intracellular Proliferation Assay of S. Typhimurium

RAW264.7 cells were cultured in 12-well culture plates at 37 °C with 5% CO_2_ overnight. Then, cells were infected with the *S.* Typhimurium (MOI 100 bacteria per cell), for 1 h, and treated with gentamicin (100 μg/mL) for 1 h. After washing, the cells were cultured in medium containing a low concentration of gentamicin (10 μg/mL) overnight, followed by enumeration of *S.* Typhimurium using the spread plate method. Bacterial survival was depicted as the ratio of invaded bacteria at 2 h.

### 4.6. Measurement of the Binding Ability of r0306

The binding ability of r0306 was detected according to the method described by Kaur et al. [14]. Briefly, IPEC-J2 cells were cultured in 96-well culture plates at 37 °C with 5% CO_2_ until they had grown to approximately 60% confluence. After being fixed in cooled 4% paraformaldehyde solution for 30 min and blocked with 5% bovine serum albumin (BSA) for 2 h, the cells were treated with r0306 or BSA for 1 h and then incubated with His-Tag antibody (Proteintech, Chicago, IL, USA) for 2 h and HRP-conjugated secondary antibody (Sangon, Shanghai, China), at 37 °C for 1 h. The binding ability of r0306 to IPEC-J2 cells was determined by OD_450_ using a microplate reader.

### 4.7. Cell Staining

IPEC-J2 cells were seeded on sterilized glass coverslips placed on 6-well culture plates at 37 °C with 5% CO_2_ until they had grown to approximately 60% confluence, r0306 treatment and bacterial infection were performed as described in the adhesion assay. After washing with PBS and fixing with methanol, the stained cells were incubated with Wright and Giemsa stain (Solarbio, Beijing, China) for 2 min. The stained cells were then observed under a microscope.

To observe the cytoskeleton, IPEC-J2 cells were treated as described above. Following incubation with r0306 and fixation, the cells were stained with FITC-labeled phalloidin (Solarbio, Beijing, China) and DAPI. The stained cells were then observed under a fluorescence microscope.

### 4.8. Fluoroscopy of Cells

To observe adhered bacteria by fluorescence microscope, we constructed a spontaneous green fluorescence-labeled strain of *S.* Typhimurium (ST-EGFP) by using plasmid pUC19-EGFP. IPEC-J2 cells were cultured in 96-well culture plates at 37 °C with 5% CO_2_ until they had grown to approximately 60% confluence, and then treated with r0306 (10 μg/mL) or eluent solvent for 1 h, followed by infection with the ST-EGFP strain. After washing and fixation, the cells were stained with DAPI (MIKX, Shenzhen, China) and observed under a fluorescence microscope.

### 4.9. Cell Viability Assay

IPEC-J2 cells were cultured in 96-well culture plates at 37 °C with 5% CO_2_ until they had grown to approximately 90% confluence, and then treated with r0306 or eluent solvent for 1 h. Afterwards, the cells were incubated with Cell Counting Kit-8 (CCK-8, MIKX, Shenzhen, China) at 37 °C for 2 h. The viability of the cells was calculated by measuring the OD_450_ with a microplate reader.

### 4.10. RNA Extraction and Quantitative Realtime PCR (qRT-PCR)

*S.* Typhimurium strains were grown in LB medium, at 37 °C and on a shaker (220 r/min), until OD_600_ reached 0.8–1.0, and the total RNA was extracted using a BIOG bacterial RNA extraction kit (BAIDAI, Changzhou, China). Cells were cultured in 12-well culture plates at 37 °C with 5% CO_2_ until they had grown to approximately 80% confluence, and treated with r0306 or eluent solvent for 10 h. The total RNA was extracted using a FastPure^®^ Cell/Tissue Total RNA isolation Kit (Vazyme, Nanjing, China). Reverse transcription of total RNA was conducted using a HiScript^®^ II Reverse Transcriptase Kit (Vazyme, Nanjing, China). Quantitative PCR was operated on a CFX96 Touch Real-Time PCR Detection System (Bio-Rad, Hercules, CA, USA) using AceQ qPCR SYBR green master mix (Vazyme, Nanjing, China). The mRNA expression of target genes in host cells was normalized to that of glyceraldehyde-3-phosphate dehydrogenase (*GAPDH*), while the mRNA expression of target genes in *S.* Typhimurium was normalized to that of DNA gyrase subunit A (*gyrA*). The results were calculated using the 2^−ΔΔCt^ method. The sequence information for all primers used in the qRT-PCR is listed in Appendix A.

### 4.11. Western Blot

IPEC-J2 cells were cultured in 6-well culture plates at 37 °C with 5% CO_2_ until they had grown to approximately 80% confluence, and treated with r0306 or eluent solvent for 10 h. After washing with PBS, the cells were lysed on ice using radio immunoprecipitation assay (RIPA) lysis buffer (Beyotime, Shanghai, China) containing phenylmethylsulfonyl fluoride (PMSF). The supernatant was collected at 4 °C and denatured at 99 °C for 10 min. Total cell protein was separated by SDS-PAGE and target proteins were detected using the rabbit anti-phospho-p65 antibody (Cell Signaling Technology, Danvers, MA, USA) and rabbit GAPDH antibody (Proteintech, Chicago, CA, USA). After visualization of the protein blots, the Gel-Pro Analyzer (Media Cybernetics, Rockville, MD, USA) was used to determine the expression of target proteins, normalized to GAPDH expression.

### 4.12. In Vivo Evaluation of the Virulence of S. Typhimurium Strains

A total of 15 5-week-old female BALB/c mice of similar body weight were randomly assigned to 3 groups, with 5 mice per group. Mice were housed in sterilized cages and maintained at a temperature of 24 degrees Celsius. Food was replaced and drinking water was changed every afternoon, and lights were on for 12 h per day. Mice fasted for 6 h and were given streptomycin by gavage (7.5 mg per mouse). One day later, after another 6 h fast, mice in each group were orally gavaged with 1 × 10^8^ CFU of either the wild-type, Δ*0306,* or CΔ*0306* strain of *S*. Typhimurium. Body weight changes in the mice were measured daily. To detect the colonization capacity of *S*. Typhimurium, fresh feces were collected after one day of infection and dissolved in PBS. Fecal *S*. Typhimurium was enumerated by the spread plate method on *Salmonella-Shigella* agar. The number of dead mice in each group was recorded daily, and a Kaplan–Meier plot of cumulative mortality was constructed to compare survival rates.

### 4.13. Statistical Analysis

All data are expressed as the mean ± standard error and were analyzed using the one-way ANOVA in SPSS 21.0. Differences between groups were examined using Duncan’s multiple comparisons. * *p* < 0.05 and ** *p* < 0.01 were considered statistically significant.

## 5. Conclusions

In summary, our study shows that STM0306, a homolog of PagN, has the ability to adhere to and invade intestinal epithelial cells, subsequently causing cellular inflammation and cytoskeleton destruction in the host. Deletion of the *STM0306* gene attenuates the pathogenicity of *S.* Typhimurium in vivo. Notably, there may be a synergy between *STM0306* and other invasion-related genes of *S.* Typhimurium, the specific mechanism of which deserves further investigation.

## Figures and Tables

**Figure 1 ijms-24-08170-f001:**
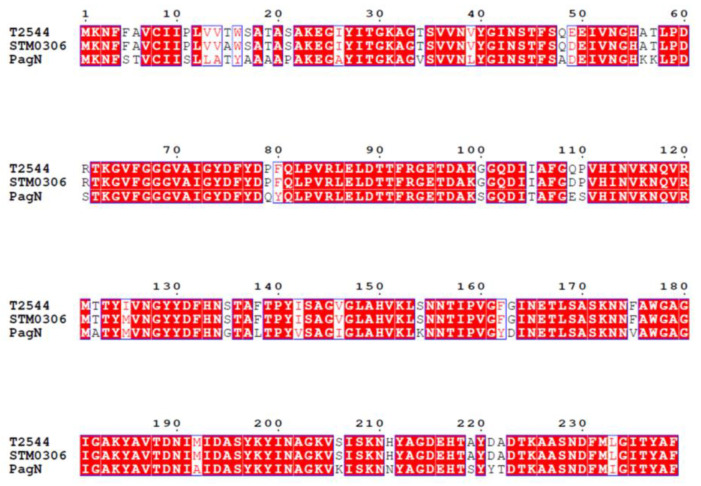
Protein sequence homology analysis of T2544 (*S*. Typhi), STM0306 (*S.* Typhimurium), and PagN (*S. bongori*) by using CLUSTALW. The red background indicates the fusion loop and the red text indicates the glycosylation side.

**Figure 2 ijms-24-08170-f002:**
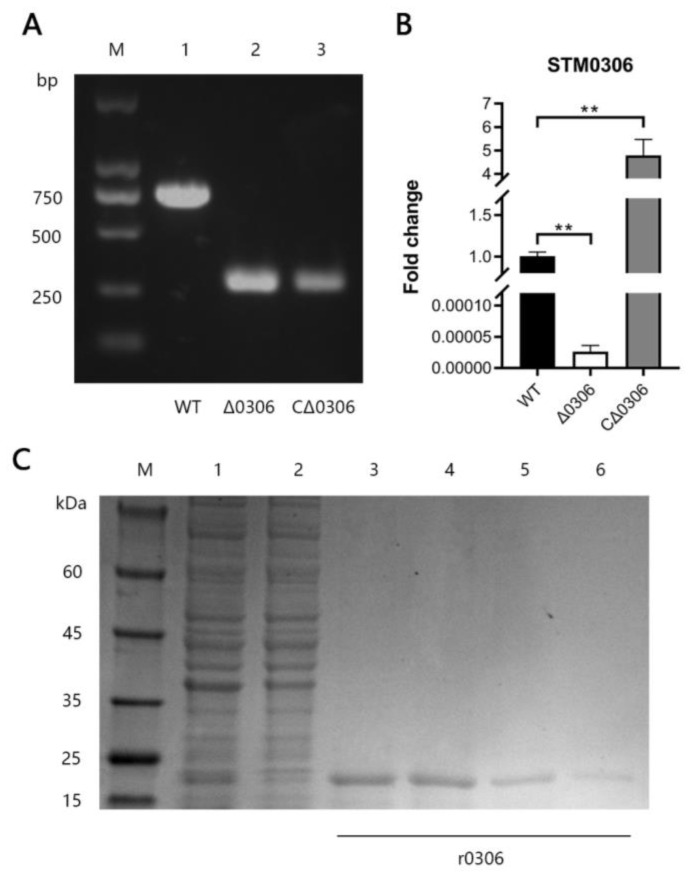
Construction of the *S.* Typhimurium *STM0306* mutant strain and purification of the recombinant STM0306 protein. (**A**) Knockout of *STM0306* by λ-red recombination system and PCR amplicon validation of Δ*0306* and CΔ*0306* strains. M: 2000 bp DNA marker; lane 1: wild-type amplification product (726 bp); lane 2: Δ*0306* strain amplification product (260 bp); lane 3: CΔ*0306* strain amplification product (260 bp). (**B**) Verification of *STM0306* mRNA expression in the Δ*0306* and CΔ*0306* strains by qRT-PCR (*n* = 6). The results are shown as the mean ± standard error, ** means *p* < 0.01. (**C**) Expression of *STM0306* gene in *E. coli* BL21 (DE3) via pET-32a containing *STM0306* gene and purification of r0306 from bacterial lysate supernatant by Ni-NTA affinity chromatography. Purity analysis of r0306 by SDS-PAGE. M: 180 kDa protein marker; lane 1: the supernatant of the bacterial solution after sonication; lane 2: the impurities from the washer; lanes 3–6: the purified r0306 proteins (24 kDa).

**Figure 3 ijms-24-08170-f003:**
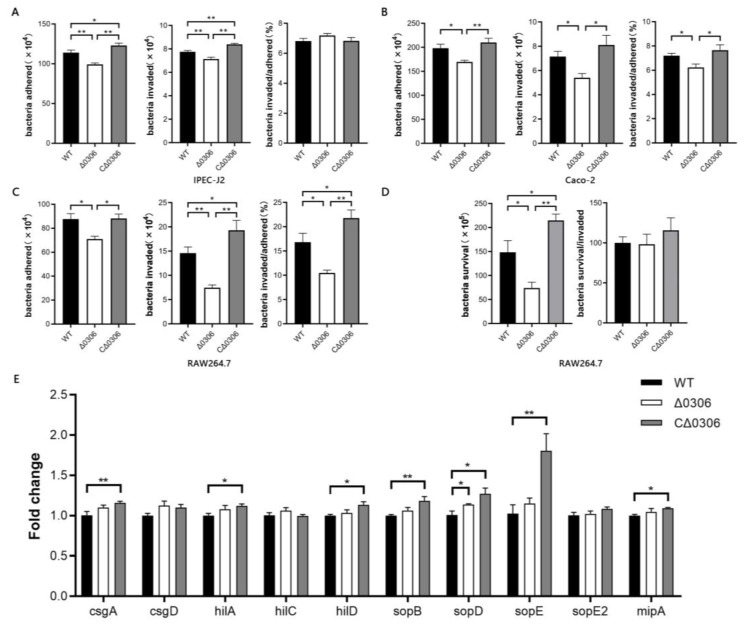
STM0306 is highly related to adhesion and invasion of *S.* Typhimurium. (**A**–**C**) The effect of *STM0306* gene deletion on adhesion and invasion of *S.* Typhimurium to IPEC-J2, Caco-2, and RAW264.7 cells (*n* = 4). Cells were infected with each of the three strains (WT strain, Δ*0306* strain, and CΔ*0306* strain) of *S.* Typhimurium. Cells were washed with PBS and lysed with PBS containing Triton X-100, followed by enumeration of *S.* Typhimurium using the spread plate method. Bacterial invasiveness was depicted as % of bacteria adhered. (**D**) Proliferation of *S.* Typhimurium in RAW264.7 cells (*n* = 4). Cells were infected with *S.* Typhimurium, and then cultured in medium containing low-concentration gentamicin after gentamicin treatment to kill the extracellular adherent bacteria. Bacterial survival is depicted as the ratio of invaded bacteria at 2 h. (**E**) Relative mRNA expression of invasion-related genes of the WT, Δ*0306*, and CΔ*0306* strains (*n* = 6). All results are shown as the mean ± standard error, * means *p* < 0.05, and ** means *p* < 0.01.

**Figure 4 ijms-24-08170-f004:**
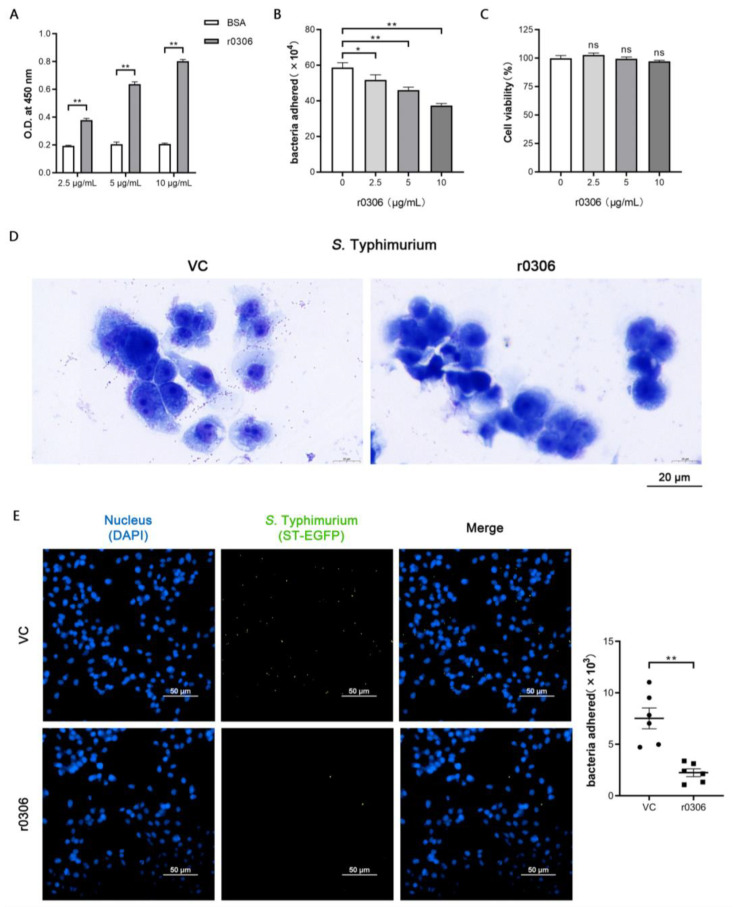
STM0306 reduces the *S.* Typhimurium adhesion via binding to host cells. (**A**) ELISA-based assay of r0306 adhesion to IPEC-J2 cells (*n* = 3). The binding ability of r0306 was determined by OD_450_ after incubation with His-Tag antibody and HRP-conjugated secondary antibody. (**B**) IPEC-J2 cells were pretreated with r0306 and a study on the effect of r0306 on the adhesion of *S.* Typhimurium to IPEC-J2 cells was performed using a standard adhesion assay (*n* = 4). (**C**) Detection of viability of IPEC-J2 cells using CCK-8 (*n* = 6). (**D**) Wright–Giemsa staining of IPEC-J2 cells after r0306 preincubation and infection with *S.* Typhimurium; blue for cells, purple for bacteria. (**E**) Fluorescence microscopy of IPEC-J2 cells after r0306 pretreatment and *S.* Typhimurium infection using the spontaneous green fluorescence strain of *S.* Typhimurium (ST-EGFP), calculation of the number of bacteria in the 96-well culture plates according to the green fluorescence in the field of vision (*n* = 6). All results are shown as the mean ± standard error, ns means no significant difference compared with the control group, * means *p* < 0.05, and ** means *p* < 0.01.

**Figure 5 ijms-24-08170-f005:**
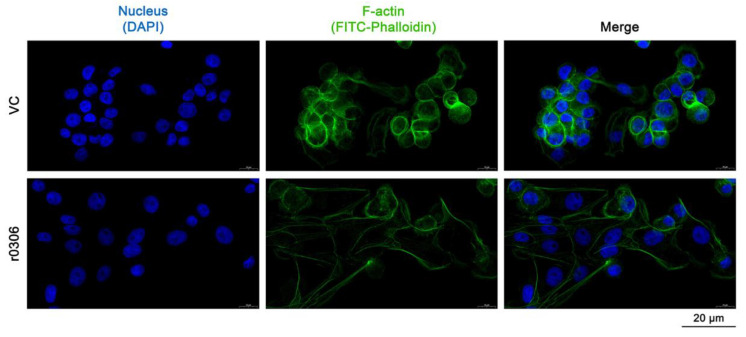
Actin modulation of IPEC-J2 cells induced by r0306. The cytoskeleton (F-actin) was stained with FITC-labeled phalloidin and the cell nucleus was stained with DAPI.

**Figure 6 ijms-24-08170-f006:**
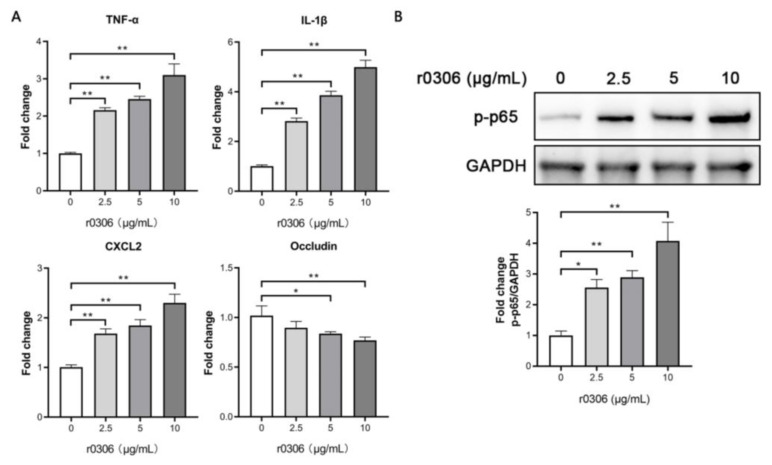
STM0306 causes host cell inflammation through the NF-κB signaling pathway. (**A**) Relative mRNA expression of inflammatory cytokine genes of IPEC-J2 (*n* = 6). (**B**) Protein expression analysis of IPEC-J2 cells by Western blot; the densitometric analysis shows the level of *p*-p65 over GAPDH (*n* = 3). All results are shown as the mean ± standard error, * means *p* < 0.05, and ** means *p* < 0.01.

**Figure 7 ijms-24-08170-f007:**
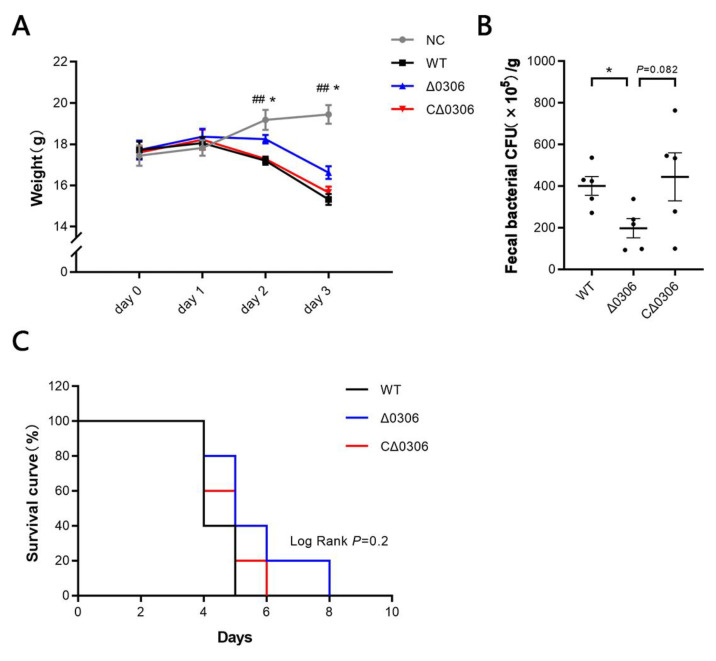
STM0306 is implicated in the pathogenicity of *S.* Typhimurium. (**A**) Daily weight changes in mice after *S.* Typhimurium infection (*n* = 5). (**B**) The number of bacteria in the feces of mice one day after *S.* Typhimurium infection (*n* = 5). (**C**) Survival curve of infected mice. After bacterial infection, the status of mice was observed every day (*n* = 5). All results are shown as the mean ± standard error, * means Δ*0306* group compared with the wild-type group *p* < 0.05, ## means wild-type group compared with the control group *p* < 0.01.

**Table 1 ijms-24-08170-t001:** Bacterial strains and plasmids used in this study.

Strains	Description	Source
*S.* Typhimurium		
LT2	Wild-type	ATCC
Δ*0306*	*STM0306* mutant strain	This study
CΔ*0306*	Δ*0306* complemented with *STM0306* gene in plasmid pBR322; Amp^r^	This study
ST-EGFP	Wild-type transformed with *EGFP* in plasmid pUC19; Amp^r^	This study
*Escherichia coli*		
DH5α	Clone strain	Laboratory stock
BL21 (DE3)	Expression strain	Laboratory stock
Plasmids		
pKD46	Amp^r^; lambda-red recombinase plasmid	Laboratory stock
pKD4	Kan^r^; template plasmid	Laboratory stock
pCP20	Amp^r^; cre-recombinase expression plasmid	Laboratory stock
pBR322	Amp^r^; clone plasmid	Laboratory stock
pUC19-EGFP	Amp^r^; EGFP recombinase plasmid	Laboratory stock

## Data Availability

All the data were published in this article.

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
