# Peer review of "A Potential Adhesin/Invasin STM0306 Participates in Host Cell Inflammation Induced by *Salmonella enterica* Serovar Typhimurium"

_ijms, 2023, doi:10.3390/ijms24098170_

Round 1
Reviewer 1 Report
This paper is well written but two major points need to be clarified before acceptance:
What is the difference between STM0306 and PagN from these papers: 10.1186/1471-2180-8-142 and Barilleau et al? The role and phylogenetic relation of PagN and other related proteins should be introduced clearly in the paper. Only three sequences do not give a clear idea of the protein of interest. Adding phylogenic analysis on more sequences would be an asset for this work.
Likewise, the authors should discuss this work: 10.3390/microorganisms8040489
More details about the conditions of animal maintenance should be stated in order to help other teams to redo the work. What about the ethical committee of this work regarding animal welfare?
A minor comment is about the adhesins of Salmonella. The flagella are well known to be involved in the adhesion and should be added in the introduction.
Reviewer 2 Report
The authors characterize STM0306, a protein with putative adhesin/invasion roles in Salmonella. Using a deletion strain of STM0306, they show that this gene is involved in (i) adhesion and invasion of IPEC-J2 and Caco-2 cells, (ii) survival inside RAW macrophages, (iii) host cell inflammation and (iv) virulence in a typhoid fever murine model. The data are solid and of general interest in the field of infection biology. However, the manuscript could be improved as many result sections and several technicalities lack a proper introduction/description, making it difficult to read and sometimes confusing for the reader. In addition, the authors point would be strengthened by generating a non-overexpressing complementation strain and by distinguishing the role of STM0306 in SPI-1 and SPI-2 regulons. Please find my comments below.
Introduction. The introduction lacks a description of PagN and T2544 and their roles in adhesion/invasion. As these proteins are often referred to, it would make the paper clearer to have a good understanding of the mechanisms of action of PagN-dependent adhesion/invasion of host cells, including the type III secretion system I, the zipper mechanism of invasion, and regulation of expression of PagN.
Lines 66-69. Please quantify % identity.
Lines 77-79. Please describe how the complementation was done? The material and methods section describes the use of a pBR322 plasmid. This explains the higher level of STM0306 than in the WT. It is also not described what promoter was used for complementation, was it the native promoter? Please clarify and discuss these aspects in this section.
In addition, in the complemented strain, STM0306 is clearly overexpressed, leading to significative (unwanted) phenotypic changes as compared to the WT for (i) adherence and invasion (figure 3A-B), (ii) fitness in RAW macrophages (figure 3D) and (iii) expression levels of invasion-related genes csgA, hilA, hilD, sopD, sopE and mipA (figure 3E). This could be avoided by generating a cleaner version of the complementation, using a low copy plasmid (pSC101) with STM0306 native promoter or by restoring the WT STM0306 locus at the chromosome with its native promoter. I suggest the authors might keep the current construct as it is useful to describe phenotypes upon overexpression of the STM0306 gene, but add a clean complementation, which should restore the WT phenotype for all tests in Figure 2B and figure 3. This should strengthen the manuscript and clarify/simplify the message as the authors currently describe “remarkable or surprising” effects in the complemented strain (see lines 77-79, 102-104, 123-124, 130-133) that likely reflect the overexpression of STM0306 rather than its complementation.
Lines 96-99. Could the authors briefly describe IPEC-J2 and Caco-2 cell lines and why this is a good model for invasion/adhesion in this paragraph?
Lines 119-124. Survival inside macrophages like RAW264.7 is mediated by SPI-2 T3SS in Salmonella. The results thus suggest a role for STM0306 not only in adhesion/invasion (SPI-1 mediated) but also in survival inside macrophages (SPI-2 mediated). This is notably in line with the mice experiment performed in figure 6. Could the authors please elaborate on this and clearly distinguish these two type III secretion systems in the manuscript. To validate the potentially dual role of STM0306 in Salmonella pathogenesis, the authors should verify expression of STM0306 in SPI-1 or SPI-2 inducing conditions. This can be done in vitro, by fusing the promoter of STM0306 to a gene encoding for GFP. It would strengthen the manuscript by validating/invalidating the role of STM0306 in SPI-1 or SPI-2 inducing conditions and thus its role in acute or systemic infections in vivo.
Lines 136-138. Please provide some additional information for the readers to understand why the nomenclature has changed from STM0306 to r0306. Please briefly describe how the protein is produced and purified in E. coli.
Lines 149-168. It is confusing that pre-incubation with r0306 inhibits adhesion of Salmonella while figure 3 shows that STM0306 plays a positive role in adhesion. Could the authors elaborate on this seemingly discrepancy to clarify the message here?
Reviewer 3 Report
Ling et al have explored the potential role of STM0306 as an adhesin facilitating the invasion of epithelial cells by S.Typhimurium. The experiments lack rigor and have not been performed with titrating doses. Some of the results do not agree with the final conclusion. Most of the differences in the phenotypes are very minor although been presented as statistically significant. My comments are as follows:
1. The authors acknowledge the study by Ghosh et al and that the difference might be due to difference in cell type. However, the authors do not use the conditions used by Ghosh et al as a control in their study. This is extremely important when presenting a study in contrast to the existing literature.
2. Throughout the manuscript the authors have not presented enough information in every figure as well as the figure legends making it extremely difficult to interpret the results.
3. Figure 1 - Colour codes are not described and % similarity and % identity information is missing. Figure 2C - What are lanes 2-6?
4. Figure 3 - Only one infection dose is used. 3A - difference in invasion is too less. Also authors have not plotted data properly. Number of bacteria invaded should be depicted as % of bacteria adhered. 3B - difference is too less. 3D - Invasion in RAW cells has not been looked at. CFU at 24h should be calculated as % of bacteria invaded at early time point like 2h. 3E - Except sopD and sopE other differences are too less.
5. Figure 4 - Assays with an unrelated bacterial protein should be done. Again dose titration has not been performed and it appears that the results are due to excess protein. Why is there a reduction in the number of bacteria adhered? 3B and 3D pictures are too small to look at.
6. Figure 5 - A lot of protein (10ug/ml) has been used. This increases to LPS contamination which influences cytokine gene expression significantly. Again titrating dose of protein would have helped. Most of the gene expression differences especially Occludin are minor. Experiments should also have been done with the WT and mutant bacteria.
7. Figure 6 - Body weight axis starts at 14g again stressing the point that the differences are too less and have been presented in a manner to look significant. Statistics are missing in 6B and 6C.
Round 2
Reviewer 1 Report
The authors have answered all my remarks.
Author Response
Dear reviewer:
We are grateful for your effort reviewing our paper and your positive feedback.
Reviewer 2 Report
The authors addressed most remarks very carefully and made the manuscript quality improve significantly. However, the authors did not address the complementation issue. The current complementation strain strongly overexpresses STM0306 as shown by the authors themselves by RT-qPCR at figure 2B. I therefore suggest the authors to rephrase the current text and interpret the data in Figure 3 as resulting from overexpression of STM0306. However, it is also crucial to generate a clean complementation mutant. This can be done by cloning the STM0306 ORF + 100-300 bp upstream (to include its native promoter) into a low copy plasmid such as a pSC101. Without this construct, this study lacks a crucial control to validate the results presented in this paper. The authors also mention their incapacity to obtain an STM0306 antibody to verify protein expression levels of STM0306 in the complemented strain. While this would certainly be a plus, this step is usually not required for assessing validity of a complement strain.
Author Response
Dear reviewer:
Thank you very much for the positive comments and constructive suggestions. We have rephrased the complementation strain text and reinterpreted the overexpression of STM0306 (lines 94-96, line 169 and lines 311-314). The absence of a clean complementation strain is indeed a shortcoming of our study. Your suggestions on how to construct a clean complementation strain are of great importance to our research, and we would like to thank you again for your guidance.
Round 3
Reviewer 2 Report
The changes made the manuscript very clear. However, it would gain a lot from generating a clean complementation mutant by confirming that the phenotypes observed are indeed specific to the gene studied. The overexpression strain is interesting from that perspective as it points in the opposite direction as the deletion mutant, partly confirming the specificity of the phenotypes observed. Could the authors please clearly discuss this aspect in the manuscript?
Author Response
Dear reviewer:
Thanks for your careful review. Based on your comments, we have added a description of the genetic phenotype specificity in the Discussion (lines 314-317).